# Assessment Tools to Examine Illness Understanding in Patients with Advanced Cancer: A Systematic Review of Randomized Clinical Trials

**DOI:** 10.3390/cancers17030385

**Published:** 2025-01-24

**Authors:** Ashlyn Tu, Allison de la Rosa, Kate Krause, Jessica H. Brown, David Hui

**Affiliations:** 1Department of Palliative, Rehabilitation and Integrative Medicine, The University of Texas MD Anderson Cancer Center, Houston, TX 77030, USA; atu@mdanderson.org; 2Center for Goal Concordant Care Research, The University of Texas MD Anderson Cancer Center, Houston, TX 77030, USA; ade1@mdanderson.org (A.d.l.R.); jabrown5@mdanderson.org (J.H.B.); 3Biomedical Library, The University of Texas MD Anderson Cancer Center, Houston, TX 77030, USA; kjkrause@mdanderson.org; 4Department of General Oncology, The University of Texas MD Anderson Cancer Center, Houston, TX 77030, USA

**Keywords:** communication, comprehension, neoplasms, patient outcome assessment, prognosis, terminally ill

## Abstract

Accurate patient illness understanding is associated with improved end-of-life outcomes. In this study, we investigated assessment tools used to examine illness understanding in randomized clinical trials involving advanced cancer patients. We identified five categories of assessment measures: prognostic awareness, health status, curability, treatment intent, and treatment risks/benefits. Of the 27 articles included in our final sample, we found significant heterogeneity in the questions, answer choices, how accuracy was defined, and the accuracy rates in all categories except for health status. This review shows the high variability present among illness understanding assessments, providing insights to improve the conceptualization and standardization of these approaches.

## 1. Introduction

Patients with advanced cancer encounter many difficult decisions as they approach the last months of life. Such decisions may involve palliative cancer treatments, life-sustaining treatments, cardiopulmonary resuscitation, and preparations of their family matters. Achieving the most appropriate form of care is dependent on accurate patient illness understanding, including knowledge of cancer stage, curability, prognosis, cancer terminality, cancer treatment options, treatment intent, and the risks and benefits associated with each course of action. Greater patient illness understanding is associated with improved outcomes at the end-of-life stage, including greater physician–patient trust, greater emotional preparation and acceptance, less invasive end-of-life treatment, lower readmissions, and reduced cost of care [1,2,3,4,5,6,7,8].

Despite the importance of illness understanding, various studies have demonstrated poor patient accuracy regarding cancer state and trajectory. A landmark paper by Weeks et al. reported that 69% of lung cancer patients and 81% of colorectal cancer patients did not understand that chemotherapy was extremely unlikely to cure their cancer [9]. A separate study by Sivendran et al. demonstrated only 51% of patients accurately reported cancer stage, 64% of patients accurately reported whether they were free of cancer, in remission, or had active cancer, and about 30% of patients were uncertain about their cancer status even when they were free of cancer or in remission [10].

Currently, studies have examined various aspects of illness understanding—prognostic awareness, curability, treatment intent, disease stage, illness terminality, and cancer knowledge. However, the specific questions being used to assess these aspects of illness understanding are inconsistent. These questions often have variable answer choices and heterogeneous definitions of accuracy, further complicating interpretation. Due to the sensitive nature of these questions, subtle differences in the ways these questions are asked may yield different findings. Further, illness understanding may fluctuate over time. A better understanding of the tools used to assess illness understanding in the literature and how they were assessed for accuracy could help inform future research on this important topic. In this study, we examined the assessment tools for illness understanding administered in randomized clinical trials (RCTs) involving patients with advanced cancer, how accuracy of illness understanding was assessed, and the level of accuracy of these tools. We focused on RCTs because we were most interested in how illness understanding was assessed as a variable (i.e., primary outcome, secondary outcome, co-variate, or baseline variable) in high quality studies, what the interventions were (e.g., communication tools, decision aids, or specialist palliative care), and if they specifically targeted illness understanding or other outcomes (e.g., quality of life).

## 2. Methods

### 2.1. Study Design

We performed a comprehensive systematic search adhering to the PRISMA checklist. We searched Ovid MEDLINE, Ovid EMBASE, and Web of Science. Databases were initially searched from inception to 28 February 2024. Search structures, subject headings, and keywords were tailored to each database by a medical research librarian (Kate J. Krause) specializing in systematic reviews, in consultation with our co-authors. Our search strategy consisted of Medical Subject Headings and text word or text phrase for (a) “patient”; (b) “cancer”; (c) “illness” or “prognosis”; (d) “understanding” or “knowledge”; (e) “assessments” or “questionnaires”; and (f) randomized clinical trials (see Appendix A for full search strings for all databases). This study has not been registered.

### 2.2. Eligibility Criteria

We included only RCTs because they provide a high level of evidence, thus providing a state of the science overview regarding how illness understanding was assessed in the contemporary literature. Moreover, studies that examined illness understanding specifically as an outcome can provide information on its responsiveness to change and inform future RCTs on communication interventions. We limited studies to those that reported on adult patients with advanced cancer. Non-English articles, case reports, conference materials, protocols, and theses were excluded. Deduplication was performed manually in EndNote. The institutional review board at MD Anderson Cancer Center waived the need for a full committee review.

### 2.3. Data Collection

Articles from the initial search were imported into the Covidence systematic review software (Veritas Health Innovation, Melbourne, Australia) for further screening. Two investigators (A.T., A.d.l.R.) independently reviewed the titles and abstracts of the articles to identify potentially relevant studies. Disagreements were resolved by consensus and by seeking the opinion of a third reviewer (D.H.). Studies that passed the title/abstract review were retrieved for full-text review. The two investigators (A.T., A.d.l.R.) then independently screened the remaining full-text articles. Disagreements were resolved by consensus and by seeking the opinion of a third reviewer (D.H.).

We collected data on manuscript characteristics, including the year of publication, country of the corresponding author, patient population (cancer diagnosis, cancer stage), sample size, primary outcome of the RCT, and main outcome of the specific manuscript.

### 2.4. Data Analysis

We assessed the quality of reporting of RCTs using the Jadad score. The Jadad score is a three-item scale based on how the article described the randomization process, double-blinding, and withdrawals/dropouts. The higher the score, the greater an indication of higher quality reporting. The lowest possible score is 0, and the highest possible score is 5. A Jadad score of 3 or greater indicates a high-quality RCT [11].

After examining the questions used in each study, we identified 5 themes and concepts related to assessing patient illness understanding: prognostic awareness, health status, curability, treatment intent, and treatment risks/benefits. Following this, we documented the questions, answer choices, and definitions of accuracy each study used to assess patient illness understanding. We also retrieved the accuracy rate of patients’ responses as defined by the authors in each study. We used descriptive statistics to summarize the data, including percentages, means, medians, and 95% confidence intervals.

## 3. Results

### 3.1. Study Characteristics

The final sample included 27 articles based on data from 16 RCTs. Figure 1 shows the study flow diagram. As shown in Table 1, a majority of these articles were published between the years 2020–2024 (*n* = 16, 59%), appeared in oncology journals (*n* = 14, 52%), were from the United States of America (*n* = 19, 70%), and had mixed cancer diagnoses (*n* = 19, 70%). Twenty-four (89%) articles included only patients with advanced cancer, while three (11.1%) also included some patients with localized disease. The median Jadad score was 2 (interquartile range 1–2). Four (15%) articles reported primary data from the RCT, while twenty-three (85%) were secondary analysis (Table 2).

### 3.2. Assessment of Patient’s Prognostic Awareness

We identified six different ways prognostic awareness was assessed among 10 articles based on eight RCTs (Table 3). Four studies asked patients about the probability of their 2-year survivability (0–100% in categories) [12,13,14,15]. Although the specific questions of the other four studies varied, they shared the common question stem asking patients to “estimate” or “guess” their “life expectancy” [16,17,18,19,20,21]. These studies asked about life expectancy through a variety of answer choices, including time frames (e.g., 6 months or less, 5 years or more), descriptive categories (e.g., months vs. years), or age of death. Only four RCTs defined accuracy based on the more pessimistic responses [15,16,17,18,19,20,21]. The accuracy rate ranged from 6% to 33%.

### 3.3. Assessment of Patient’s Understading of Health Status

Health status was assessed in four articles from three RCTs (Table 3). All questions asked patients to describe their “current health status” with similar categories of descriptors (relatively healthy, seriously ill, and/or terminally ill) based on the Prigerson study [16,17,22,23]. They all defined accuracy as patients’ acknowledgement that they were “terminally ill” [16,17,22,23]. The accuracy rate ranged from 45% to 59% in these studies.

### 3.4. Assessment of Patient’s Understanding of Curability

Nine articles from six RCTs assessed the patients’ understanding of curability using five different approaches (Table 3). Two studies simply asked patients to “respond to the statement “My cancer is curable” and provide a binary answer choice (yes/no) [24,25]. The question stems of the other studies had subtle differences in wording, asking patients to report what they “knew”, “perceived”, or asked them to “estimate” their curability [13,22,26,27,28,29]. The answer choices also varied widely, including descriptive categories of curability (curable, might recur, or could not be cured), descriptive categories of the chance for a cure (no chance to extremely likely), and numeric categories of a chance for a cure (0% to 100%) [13,22,24,25,26,27,28,29,30]. The definition of accuracy was provided in 5 of 6 assessments and typically required the patient to acknowledge their cancer was not curable or was unlikely to be cured. The accuracy rate ranged from 35% to 84%.

### 3.5. Assessment of Patient’s Understanding of Treatment Intent

Among the 11 articles from nine RCTs that surveyed understanding of treatment intent, nine different approaches were identified (Table 3). Specifically, four approaches asked patients if they believed that “cur[ing]” or “get[ting] rid of all [their] cancer” was a “goal of… therapy” accompanied by a binary answer choice (yes/no) [16,17,21,25,31]. Two studies asked questions related to estimations, asking patients about “the chances of” or “how likely” they thought treatment would cure their cancer [25,32,33]. Of these two studies, one contained answer choices with descriptive categories of the chance that treatment would cure the patient’s cancer (very likely to a little likely) and one contained numeric categories of the chance treatment would cure the patient’s cancer (0% to 100%) [19,32,33]. Two studies asked patients about the “primary goal” or “focus” of their current cancer treatment and used answer choices containing descriptive goals of therapy (e.g., lessen suffering, cure my cancer, extending life, help cancer research) [34,35]. Another study asked patients about the “goal(s) of chemotherapy according to their doctor” and provided descriptive goals of therapy: “cure, control cancer, improve symptoms, prolong life or other” [32]. Seven RCTs defined accuracy, which was typically based on understanding that curing cancer was not a goal of treatment [16,17,19,21,25,31,32,33,34]. The accuracy rate ranged from 26% to 88%. While Leighl et al. did not specify the exact questions and answer choices used to assess patient understanding, the paper did report that 57% of the surveyed patients understood the palliative intent of their chemotherapy, which increased in both experiment arms post consultation [12].

### 3.6. Assessment of Patient’s Understanding of Treatment Risks/Benefits

There were three articles from three RCTs that examined patient understanding of treatment risks/benefits (Table 3). One article exclusively assessed patient understanding with statements accompanied by binary answer choices of correctness (yes/no) [31]. These statements surveyed knowledge of chemotherapy symptoms (fever, diarrhea, skin ulcers), whether chemotherapy “helps people live longer” and “helps… overcome [disease] symptoms,” and how the patient’s condition state would be affected “without chemotherapy”. Accuracy was defined as choosing the true binary choice but was ultimately not reported in this paper. Another article asked patients to report “how likely” chemotherapy was to “control cancer growth” and “cause” various symptoms (nausea/vomiting, diarrhea, hair loss) [32]. However, the answer choices were not stated and accuracy was not defined. At the 2 weeks post-decision period, the intervention group had an accuracy rate of 55% and the control group had a 40% accuracy rate. Leighl et al. did not specify the exact questions and answer choices used to assess patient understanding, but it did report that at 1 to 2 weeks post consultation, patient understanding in the decision aid group improved significantly more than patients in the control group (+16% [mean +2.6 of 16 items correct] *v* + 5% [mean +0.8 of 16 items correct]; *p* < 0.001) [12]. Areas of knowledge gain in both groups included the palliative goals of therapy (+28% in DA arm *v* +13% in standard arm; *p* < 0.001) [36]. The accuracy rate ranged from 17% to 75%. A summary of the findings for each of the five assessment measurement categories is shown in Table 4.

**Table 3 cancers-17-00385-t003:** Questions to assess illness understanding from the 27 RCTs categorized by theme.

Article	Questions	Definition of Accuracy	Accuracy Rate
**Prognostic Awareness**			
**Gramling JAMA Oncol 2016** **[14]**	Patients were asked, “What do you believe are the chances that you will live for 2 years or more?” and their oncologists were asked, “What do you believe are the chances that this patient will live for 2 years or more?”… us[ing] the following response options[:]…100%, about 90%, about 75%, about 50%/50%, about 25%, about 10%, and 0%.	NR (only reported concordance)	NR (only reported concordance)
**Trevino Cancer 2019** **[20]**	Caregivers and patients were asked “What do you believe are the chances that you [the patient] will live for 2 years or more?” Response options included “100%, about 90%, about 75%, 50%/50%, about 25%, about 10%, and 0%”.	NR (only reported concordance)	NR (only reported concordance)
**Malhotra Cancer 2019** **[21]**	Patients were asked at baseline, “What do you believe are the chances that you will live for 2 years or more?”; their caregivers were asked, “What do you believe are the chances that (he/she) will live for 2 years or more?”; and their oncologists were asked, “What do you believe are the chances that this patient will live for 2 years or more?”… Options included 0%, about 10%, about 25%, 50%/50%, about 75%, about 90%, or 100%.	Accurate = “a computed index of prognostic accuracy by subtracting the 2-year prognosis as predicted by patients, caregivers, and oncologists from the vital status at 2 years (whether the patient lived for 2 years; yes = 1 or no = 0). A difference… from >−0.5 to <0.5 points was defined as realistic”.	38% of total patients were reported ‘realistic’. Regarding the estimated likelihood of 2-year survival surveyed at baseline, 36% of patients answered “100%,” 11% answered “about 90%,” 13% answered “about 75%,” 29% answered “about 50%/50% (or “don’t know”),” 5% answered “about 25%,” 3% answered “about 10%,” and 3% answered “0%”.
**Sigler JCO Oncol Pract 2022 [31]** **Loucka Palliat Med 2024 [38]**	Patients were asked “When you think about how having cancer might affect your life expectancy, do you think in terms of: months, years, or do not know?”	Accurate = months	12.8% (intervention)/11.4% (control)
**Loh Cancer 2020 [25]** **Loh JAMA Netw Open 2022** **[32]**	Patients, oncologists, and or caregivers were asked “Considering your (the patient’s) health and your (the patient’s) underlying medical conditions, what would you estimate your (the patient’s) overall life expectancy to be?” Response options were 6 months or less, 7 to 12 months, 1 to 2 years, 2 to 5 years, and more than 5 years.	Accurate = anything less than 5 years “Response of life expectancy more than 5 years was considered poor prognostic understanding regarding life expectancy estimates. We chose this most conservative definition because the study population of older adults had a variety of advanced cancers”.	NR (only reported concordance)
**Enzinger JPSM 2021** **[29]**	Patients were asked “Every person is different and their situation is unique. If you had to make a guess—based on what you have learned about your cancer, your cancer treatment, and what you know about yourself—how long do you think that you have to live?” with response options of less than one year, more than one but less than two years, more than two but less than three years, more than three but less than five years, more than five but less than ten years, and more than ten years.	Accurate = developed algorithms that designated participant’s estimate as “realistic” if 25–75% of patients in their situation would be expected to survive that long	33% (intervention)/31% (control)
**Finkelstein J Health Psychol 2022** **[33]**	Patients were asked “Using the scale below, indicate on the line how old you think you might be when you die”.	Accurate = an age within 1 year or less from their age at the time of the survey	“Only 11 (11%) and 6 (6%) participants stated an expected survival of 1 year or less in the control and intervention arms, respectively”.
**Epstein JAMA Oncol 2017** **[16]**	Physicians and patients were… asked to estimate 2-year survival … of the patients’ cancer on a 7-point scale (100%, about 90%, about 75%, about 50/50, about 25%, about 10%, 0%, don’t know).	NR (only reported concordance)	NR (only reported concordance)
**Health Status**			
**Sigler JCO Oncol Pract 2022** [31] **Loucka Palliat Med 2024** [38]	Patients were asked “How would you describe your current health status: relatively healthy but terminally ill, seriously and terminally ill, relatively healthy, or seriously but not terminally ill”.	Accurate = ‘relatively healthy but terminally ill’, ‘seriously and terminally ill’ (acknowledged that they were terminally ill)	54% (intervention)/45% (control)
**Gray Palliat Med 2023** [35]	Used items from the PTPQ that asked “patient[s] to describe their current health status at baseline… week-12, and week-24… by choosing from the following mutually exclusive options: ‘relatively healthy’, ‘relatively healthy and terminally ill’, ‘seriously ill and not terminally ill’, or ‘seriously ill and terminally ill’”.	Accurate = ‘seriously ill and terminally ill’ and ‘relatively healthy and terminally ill’	“Among the total cohort of 350 patients, 58% reported that they were terminally ill at baseline. From baseline to week-24, there were changes in patients’ perceptions of terminal illness, in which 59% of patients either remained or became accurate in their acknowledgement of terminal illness, whereas 41% either remained or became inaccurate at week-24”. More specifically, 36% remained accurate and 22% became accurate.
**Emanuel J Palliat Med 2023 [36]**	Asked patients “How would you describe your current health status? Response options included (1) ‘Relatively healthy’, (2) ‘Seriously but not terminally ill’, and (3) ‘Seriously and terminally ill.’”	NR	NR In this study, all patients underwent the intervention. Although the authors did not code for accuracy, they did report that 12% of the patient population had stage 1 cancer, 9% had stage 2 cancer, 18% had stage 3 cancer, and 62% had stage 4 cancer. “40% of the sample considered themselves relatively healthy, 34% reported being seriously but not terminally ill, and 25% reported being seriously and terminally ill. 11% of the participants with stage 1–3 disease described themselves as relatively healthy compared with 24% of those with stage 4 disease”.
**Curability**			
**Chen J Pain Symptom Manage (JPSM) 2019 [19]** **Wen JPSM 2019 [18]** **Chen JPSM 2020 [23]** **Wen J Natl Compr Canc Netw (JNCCN) 2020 [24]**	Patients were asked “whether they knew their prognosis, and if so, whether their disease was (1) curable; (2) might recur in the future, but their life was not currently in danger; or (3) could not be cured and they would probably die soon”.	Accurate = Option 3	At baseline: 57% (intervention)/49% (control) [19] 1–30 days before death: 76% (intervention)/68% (control) 31–60 days before death: 71% (intervention)/61% (control) 61–90 days before death: 68% (intervention)/58% (control) 91–120 days before death: 74% (intervention)/59% (control) 121–150 days before death: 66% (intervention)/56% (control) 151–180 days before death: 62% (intervention) 63% (control) [15]
**Epstein JAMA Oncol 2017 [16]**	Physicians and patients were… asked to estimate … [the] curability of the patients’ cancer on a 7-point scale (100%, about 90%, about 75%, about 50/50, about 25%, about 10%, 0%, don’t know).	NR (only reported concordance)	NR (only reported concordance)
**Gray Palliat Med 2023 [35]**	Items from the Prognosis and treatment Perception Questionnaire (PTPQ) [were used to ask]… patients to report to what extent they perceived their cancer as curable at week-12 and week-24 after study enrollment on a seven-point Likert scale (ranging from ‘no chance/0% chance’, ‘very unlikely/<10% chance’, ‘unlikely/10–24% chance’, ‘somewhat/moderately likely chance/25–74%’, ‘very likely/75–90% chance’, to ‘extremely likely/>90% chance’).	Accurate = answers ranging from ‘unlikely chance’ to ‘no chance’	“From baseline to week-24… 35% of participants either remained or became accurate in their perception that the cancer was incurable at week-24, whereas 65% of participants wither remained or became inaccurate”. More specifically, 21% remained accurate and 15% became accurate. “There was no notable differences in perception of prognosis between the Gi and lung cancer cohort”.
**Saracino JPSM 2021 [30]**	Patients were asked: “How would you describe your current disease status?” with response options[:] curable, likely curable, unlikely curable, and incurable.	Accurate = unlikely curable or incurable	63% (baseline data; total patient population)
**Greer JNCCN 2022 [34]**	Patients were asked to respond to the statement “My cancer is curable (yes/no)”.	Accurate = No	84% (intervention)/82% (control)
**Temel J Clin Oncol 2011 [13]**	Patients were asked to respond to the statement “My cancer is curable (yes/no)”.	Accurate = No	83% (intervention)/60% (control) “During the 6-month study period, a greater percentage of patients assigned to early palliative care either remained or became accurate in the perception that the cancer was not curable compared with patients receiving standard care”.
**Treatment Intent**			
**Sigler JCO Oncol Pract 2022 [31]** **Loucka Palliat Med 2024 [38]**	Patients were asked to respond to the statement “the goals of my therapy are to get rid of all my cancer (Yes/No)”.	Accurate = No	26% (intervention)/ 36% (control)
**Nipp Ann Oncol 2016 [15]**	Patients were asked at baseline “to choose the primary goal of their current cancer treatment from the options: ‘to lessen suffering’, ‘to be able to keep hoping’, ‘to make sure I have done everything’, ‘to extend my life as long as possible’, ‘to cure my cancer’, ‘to help cancer research’, and ‘other’”.	Accurate = Any response other than ’to cure my cancer’	67% of the total enrolled patient cohort (baseline data; total patient population)
**Enzinger JAMA Oncol 2020 [26]**	Patients were asked “How likely do you think that chemotherapy is to cure your cancer?” (with answer options of very likely, somewhat likely, a little likely, or don’t know). … Patients were [also] asked about the goal(s) of chemotherapy according to their doctor: cure, control cancer, improve symptoms, prolong life, or other.	1. Accurate = not at all likely 2. Accurate = control cancer, and/or improve symptoms, and or prolong life	1. 3 month follow up: 56% (intervention)/57% (control) 2 weeks post decision: 49% (intervention)/55% (control) 2. 2 weeks post decision: 88% (intervention)/87% (control)
**Finkelstein J Health Psychol 2022 [33]**	Patients were asked “Do you believe that your current treatment regimen will cure your illness?” Response options included Yes, No, and Don’t know.	Accurate = No	33% (intervention)/30% (control)
**Modes JPSM 2020 [27]**	“Which of the following best describes the focus of the medical care you are currently receiving?”… Patients could choose one of the two options (i.e., extending life… or relief of pain and discomfort…) or “I don’t know/not sure”.	NR (only reported concordance)	NR (only reported concordance)
**AlSagheir Saudi Med J 2020 [28]**	“Finally, the goal was to understand that the disease is incurable; chemotherapy helps full recovery from cancer …with response options of ‘Yes, No, and I don’t know’”.	Accurate = No	At baseline: 5% (intervention)/8% (control)
**Temel J Clin Oncol 2011 [13]**	Patients were asked to respond to the statement “The goals of my therapy are to ‘help me live longer’, (yes/no); ‘try to make me feel better’ (yes/no), or ‘get rid of all my cancer’ (yes/no)”.	Accurate = Belief that a goal of therapy was NOT to get rid of all cancer	44% (intervention)/32% (control) “Although more patients assigned to early palliative care versus those receiving standard care remained or became accurate in their belief that a goal of therapy was not to get rid of all cancer, this difference failed to meet the threshold for statistical significance”.
**Loh Oncologist 2019 [22]** **Loh JAMA Netw Open 2022 [32]**	Patients, oncologists, and/or caregivers “completed assessments of their beliefs about the curability of the cancer: ‘what do you believe are the chances the cancer will go away and never come back with treatment?’ Response options were 100%, more than 50%, 50%, less than 50%, 0%, or uncertain”.	Accurate = 0%	NR (only reported concordance)
**Leighl J Clin Oncol 2011 [12]**	This paper reported 10 items adapted from Fiset and Brundage that were used to assess patient understanding, but the exact questions asked were not reported.	NR	The paper reported that “there were no significant differences between the [intervention and control] groups immediately post consultation… most [patients] were uncertain about survival outcomes; 36% of all patients were familiar with survival outcomes with the addition of chemotherapy, and 23% were familiar with outcomes without chemotherapy. More than half (57%) understood the palliative intent of systemic chemotherapy in their situation”. It was also reported that “at 1 to 2 weeks post consultation, patient understanding increased in both groups, although substantially more for those randomly assigned to receive the DA (+16% [mean +2.6 of 16 items correct] *v* + 5% [mean +0.8 of 16 items correct]; *p* < 0.001). Areas of knowledge gain in both groups included the palliative goals of therapy (+28% in DA arm *v* +13% in standard arm; *p* < 0.001) and better understanding of survival outcomes with chemotherapy… Understanding of the risk of severe chemotherapy toxicity declined during the 1-to-2-week interval post consultation in both groups”.
**Treatment Risks/Benefits**			
**AlSagheir Saudi Med J 2020 [28]**	1. “First, chemotherapy-induced toxicity… includes fever (more than 38 degrees) and severe diarrhea (more than 6 times above normal) … Response options included ‘Yes, No, and I do not know.’” 2. “Chemotherapy-induced toxicity… includes… painful numbness in the limbs and skin ulcers in the feet and hands (allergies)”. Response options included ‘Yes, No, and I do not know’”. 3. “The second outcome with chemotherapy helps people live longer, and chemotherapy helps you overcome some of the symptoms and problems you experience. Response options included ‘Yes, No, and I do not know’”. 4. “‘Third outcome without chemotherapy, which includes the following: my condition will worsen without chemotherapy.’ Response options included ‘Yes, No, and I do not know.’”	1. Accurate = Yes 2. Accurate = No 3. Accurate = Yes 4. Accurate = Yes	At baseline: 1. and 2. (combined): 58% (intervention)/54% (control) 3. 65% (intervention)/74% (control) 4. 75% (intervention)/65% (control)
**Enzinger JAMA Oncol 2020 [26]**	Patients also reported how likely chemotherapy was to control cancer growth and cause nausea/vomiting, diarrhea, neuropathy, and hair loss	NR (Answers were not dichotomized into accurate/inaccurate given the subjectivity of the item wording.)	2 weeks post decision: 55% (intervention)/40% (control)
**Leighl J Clin Oncol 2011 [12]**	This paper reported 10 items adapted from Fiset and Brundage that were used to assess patient understanding, but the exact questions asked were not reported.	NR	The paper reported “no significant differences between the [intervention and control groups] immediately post consultation… 75% of patients understood the impact of chemotherapy and adverse effects. 17% of patients understood the risks of developing severe (grade >/= 3) of chemotherapy toxicity, with most overestimating this risk”. It was also reported that “at 1 to 2 weeks post consultation, patient understanding increased in both groups, although substantially more for those randomly assigned to receive the DA (+16% [mean +2.6 of 16 items correct] *v* + 5% [mean +0.8 of 16 items correct]; *p* < 0.001)… Understanding of the risk of severe chemotherapy toxicity declined during the 1-to-2-week interval post consultation in both groups”.
**Other**			
**deRooij Qual Life Res 2018 [17]**	“The Brief Illness Perception Questionnaire (B-IPQ) was used to assess illness perceptions after initial treatment. The B-IPQ includes 8 single-item scales… only the scales that have earlier shown to be affects by SCPs in our trial were used in the analysis, including the amount of symptoms experiences, concerns about the illness, emotional impact of the illness with respect to endometrial cancer, and trust that the treatment would help to cure with respect to ovarian cancer”. While the specific questions asked were not explicitly stated in the manuscript, upon further investigation of the B-IPQ, we extracted the following questions as relevant (all responses were on a scale of 1–10): 1. How much do you experience symptoms from your illness? (0 (no symptoms at all)–10 (many severe symptoms)) 2. How concerned are you about your illness? (0 (not at all concerned)–10 (extremely concerned)) 3. How much does your illness affect you emotionally? (e.g., does it make you angry, scared, upset, or depressed?) (0 (not at all affected emotionally)–10 (extremely affected emotionally)) 4. How much do you think your treatment can help your illness? (0 (not at all)–10 (extremely helpful))	NR	NR
**Guan Psychooncology 2023 [37]**	Used “a researcher-modified shortened version of the Mishel Uncertainty in Illness Scale for Adult (MUIS-A). The modified scale included 9 items with a 4 point Likert scale ranging from 1 (not at all) to 4 (a lot)”. The specific questions patients were asked were not given.	NR; “Total possible scores ranged from 9 to 36, with higher scores indicating greater levels of uncertainty”.	NR; Only the mean score of patient illness uncertainty was provided (20.42)

Abbreviations: NR—Not Reported.

**Table 4 cancers-17-00385-t004:** Summary of findings for each of the five assessment measurement categories.

Assessment Measurement Category	Number of Articles/RCTs	Number of Unique Assessments	Accuracy Rate (Range)
**Prognostic Awareness**	10 articles, 8 RCTs	6	6% to 33%
**Health Status**	4 articles, 3 RCTs	1 overall main question, but answer choices varied slightly	45% to 59%
**Curability**	9 articles, 6 RCTs	5	35% to 84%
**Treatment Intent**	11 articles, 9 RCTs	9	26% to 88%
**Treatment Risks/Benefits**	3 articles, 3 RCTs	3	17% to 75%

## 4. Discussion

This study examined the existing approaches to assess illness understanding in 27 articles and classified them under five key themes: prognostic awareness, health status, curability, treatment intent, and treatment risks and benefits. Prognostic awareness and treatment intent were most frequently assessed. Apart from questions assessing health status, we identified wide variations in assessment questions, answer choices, how accuracy was defined, and accuracy rates. By highlighting the strengths and weaknesses of each approach, this study may inform how illness understanding can be assessed both clinically and in research.

Illness understanding is poorly defined, making it particularly challenging to conduct a literature review [39]. Indeed, in an integrative review, Finlayson et al. conceptualized illness understanding using five categories that were similar to the ones in our review: understanding of cancer diagnosis, knowledge of prognosis, combinations of knowledge of diagnosis and prognosis, understanding the disease trajectory, and understanding of curability and survivability. We decided to keep treatment intent and awareness of curability as separate categories despite the significant overlap because previous studies have identified significant discordance in these two concepts [13,39], with treatment intent focusing on the goal of cancer therapy and awareness of curability assessing what the patient believes is possible with their cancer.

The assessment questions tended to ask patients what they “think/estimate/believe/feel” rather than what they were explained or told by their oncologist [3,9,10,14,15,19,20,23,25,26,27,32], which may yield different answers. For patients to report their illness understanding accurately, they need to be provided accurate medical information, be able to comprehend this information intellectually, and be able to accept the information emotionally. Additionally, patients need to be able to articulate this information and respond to the survey appropriately, provided that the questions and answers were clearly stated and understandable. This may explain why the accuracy rates were relatively low. Further studies may consider examining different aspects (what patients recalled they were told vs. what patients perceived that information to mean vs. what patients hoped for) to better identify the gaps in illness understanding.

Conceptualizing and operationalizing accuracy is challenging given the wide variety of questions and answer options. For example, some answers were compounded (e.g., ‘unlikely/10–24% chance of curability’), which could potentially create confusion for respondents if they had a different understanding of what an “unlikely chance” of curability means [29]. Some studies only considered patients to have accurate illness understanding if they selected “no probability of cure”, while other studies would consider “unlikely” to be an acceptable indicator of accurate understanding [2,24,28,29]. The heterogeneous assessment methods, coupled with different patient populations and study design, likely contributed to the wide range of accuracy observed for prognostic awareness (6% to 33%), curability (35% to 84%), and treatment intent (26% to 88%). In contrast, health status was relatively consistent (45% to 59%), likely because of the consistent approach in assessing this aspect of illness understanding.

Only four studies in our sample focused on illness understanding as a primary outcome. Leighl et al. reported that a chemotherapy decision aid intervention significantly improved patient understanding 1 to 2 weeks post consultation [12]. Enzinger et al. found the palliative care educational video and booklet did not significantly improve patients’ understanding of treatment intent and treatment risks/benefits [23]. Al Sagheir et al. reported that their decision aid intervention did not improve patients’ prognostic understanding [22]. The Finkelstein study found that using an accuracy incentivized survey and attribute framing tests did not significantly improve prognostic awareness or understanding of treatment intent [27]. Given that illness understanding is a key step to delivering goal concordant care, it is important to develop better interventions and assessment tools to improve patient-clinician communication and illness understanding.

This study has several limitations. First, we only included 27 articles, which may not represent all the possible questions used to assess illness understanding. Second, we only focused on RCTs because we aimed to identify how experimenters assess illness understanding in interventional studies. Because our study focused on RCTs, which are typically performed at academic centers that serve a different patient population compared to smaller hospitals, further studies are needed to examine patient illness understanding in other settings. Third, we also did not search the Grey Literature database and included English-only articles. A majority of the included studies were conducted in North America (*n* = 20, 74%). Because cultural factors may influence patients’ illness understanding and how it should be assessed, inclusion of studies in other languages may provide a more global perspective. Fourth, we were unable to examine the mode of survey administration because the survey administrator (i.e., the physician, research staff, patient) was not always clearly reported. Fifth, because illness understanding fluctuates, longitudinal assessments are needed. Our systematic review is limited to what was reported and could not assess this in detail. However, these fluctuations may make it more difficult to detect signals in studies aiming to improve illness understanding and need to be further studied. Lastly, the majority of these studies examined illness understanding at baseline only and reported the findings as secondary analyses instead of being the primary outcome for a communication intervention. Given the high level of inaccuracy in illness understanding even at baseline, our study highlights the opportunity for systematic and routine screening in clinical settings to identify gaps in illness understanding and need for psycho-social-spiritual support.

Our findings have potential implications for clinical practice and research. Given that curability (yes/no) and treatment intent (curative/palliative) are typically binary in nature, it may make sense to use simple questions with binary choice for patients to assess these constructs. Probability-based questions are also harder to code as accurate or inaccurate. In addition, such questions may be interpreted differently depending on each individual’s outlook (i.e., positive or negative) and numeracy. Currently, the health status questions examined are relatively standardized. Prognosis is recommended to be communicated, in general, in time frames (months or years) instead of exact duration or probability. Thus, questions that assess patients’ prognostic understanding using similar language may be appropriate. More research is needed to examine how to assess other constructs. Reviewers and editors of such tools should also ensure investigators report their questions/answers/accuracy coding completely for accountability and reproducibility purposes. Further testing of an assessment tool’s responsiveness to change would allow us to use it as an outcome measure to assess communication and/or decision aid interventions.

## 5. Conclusions

Patient illness understanding is critical to informed decision making throughout the treatment process. This study highlights the diversity of approaches to assess illness understanding in RCTs. By systematically mapping these assessments, we were able to identify opportunities to gain further insights into patients’ level of awareness of their illness, which may pave the way for clinicians to better document patients’ illness understanding and for researchers to assess interventions aimed at improving communication outcomes.

## Figures and Tables

**Figure 1 cancers-17-00385-f001:**
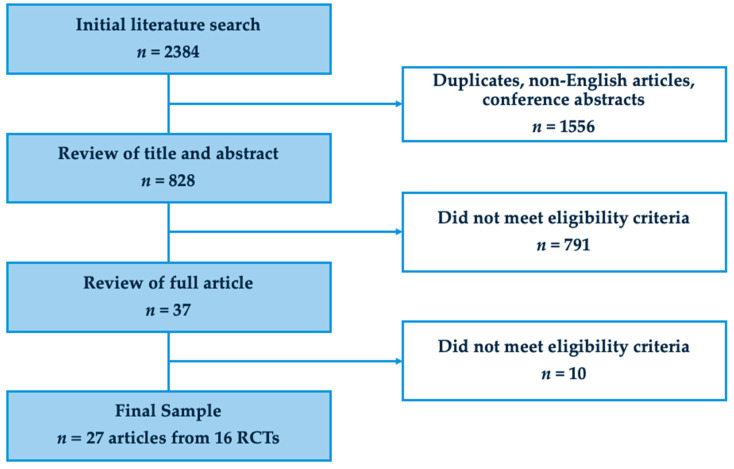
PRISMA flowchart.

**Table 1 cancers-17-00385-t001:** Article characteristics.

Variables	*n* (%)
**Year of Publication**	
2010–2014	2 (7)
2015–2019	9 (33)
2020–2024	16 (59)
**Journal Type**	
Oncology	14 (52)
Palliative Care	9 (33)
Others	4 (15)
**Continent of Origin**	
United States of America	19 (70)
Taiwan	4 (15)
Singapore	1 (4)
Saudi Arabia	1 (4)
Canada	1 (4)
Netherlands	1 (4)
**Secondary Analyses**	24 (89)
**Patient Population**	
Mixed cancer type	19 (70)
Gastrointestinal cancer	4 (15)
Breast cancer	1 (4)
Lung cancer	1 (4)
Gynecological cancer	1 (4)
Hematological cancer	1 (4)
**Patient Population**	
Advanced cancer only	24 (89)
Advanced and localized cancer	3 (11)
**Jadad Score (0–5)**	
1	11 (41)
2	11 (41)
3	5 (19)

**Table 2 cancers-17-00385-t002:** Included randomized clinical trials.

Author, Year	Intervention	Control	Sample Size	Primary Outcome of Original Study	Primary Outcome of Manuscript	Jadad Score
**Leighl 2011 [12]**	Consultation using decision aid for chemotherapy	Usual care	207	Prognostic understanding	Prognostic understanding and patient satisfaction regarding decision making	3
**Temel 2011 [13]**	Early palliative care	Usual care	151	Quality of life	Prognostic understanding	2
**Gramling 2016 [14]**	Multimodal intervention for high quality physician–patient communication	Usual care	236	Physician–caregiver–patient centered communication	Prognostic discordance	1
**Nipp 2016 [15]**	Early palliative care and oncology care	Usual care	350	Quality of life	Anxiety and depression	1
**Epstein 2017 [16]**	Communication intervention for oncologists, patients, and caregivers	Usual care	265	Physician–caregiver–patient centered communication	Patient–physician agreement	2
**deRooij 2018 [17]**	Survivorship care plan	Usual care	395	Supportive Care	Quality of life	2
**Wen 2019 [18]**	Interactive, individualized advanced care planning intervention	Symptom management education	460	End-of-life care treatment	End-of-life care discussions and agreement	2
**Chen 2019 [19]**	Interactive, individualized advanced care planning intervention	Symptom management education	460	End-of-life care treatment	End-of-life care discussions and agreement	2
**Trevino 2019 [20]**	Communication intervention for oncologists, patients, and caregivers	Usual care	141 (dyads)	Physician–caregiver–patient centered communication	Patient–caregiver agreement	1
**Malhotra 2019 [21]**	Communication intervention for oncologists, patients, and caregivers	Usual care	265	Physician–caregiver–patient centered communication	Prognostic accuracy	1
**Loh 2019 [22]**	Geriatric assessment summary and recommendations provided	Usual care	336 (dyads)	Patient centered communication	Quality of life, communication	1
**Chen 2020 [23]**	Interactive, individualized advanced care planning intervention	Symptom management education	460	End-of-life care treatment	Transition toward prognostic awareness	2
**Wen 2020 [24]**	Interactive, individualized advanced care planning intervention	Symptom management education	460	End-of-life care treatment	End of life care discussions and agreement	2
**Loh 2020 [25]**	Geriatric assessment summary and recommendations provided	Usual care	354 (dyads)	Patient centered communication	Patients and caregiver prognostic agreement	1
**Enzinger 2020 [26]**	Palliative care educational video and booklet at treatment initiation	Usual care	186	Patient understanding of chemotherapy benefits	Patient understanding of chemotherapy adverse effects	2
**Modes 2020 [27]**	Physicians received estimates of patient survival, CPR outcomes, and functional disability; trained nurse communicated with patient and patient’s family	Usual care	405	Patient–physician trust	Goal concordant care	1
**AlSagheir 2020 [28]**	Arabic decision aid	Usual care	92	Patient illness understanding	Patient illness understanding	3
**Enzinger 2021 [29]**	Palliative care educational video and booklet at treatment initiation	Usual care	200	Patient understanding of chemotherapy benefits	Prognostic understanding	2
**Saracino 2021 [30]**	Psychotherapies	Usual care	206	Quality of life	Patient health information preferences	1
**Sigler 2022 [31]**	Primary palliative care intervention	Usual care	457	Quality of life	Illness expectation	2
**Loh 2022 [32]**	Geriatric assessment summary and recommendations provided	Usual care	541	Patient centered communication	Patients and oncologist prognostic agreement	1
**Finkelstein 2022 [33]**	Survey that incorporated an accuracy incentive strategy and an attribute framing test	Normal survey	200	Prognostic understanding, Curability	Prognostic understanding, Curability	3
**Greer 2022 [34]**	Palliative care intervention for end-of-life care discussions	Usual care	120	Documentation of end of life care discussions	Patient-reported discussions about end-of-life care preferences	3
**Gray 2023 [35]**	Early palliative care and oncology care	Usual care	350	Quality of life	Perceptions of prognosis	3
**Emanuel 2023 [36]**	Dignity therapy	Usual care	452	Death anxiety/distress	Death anxiety/distress	1
**Guan 2023 [37]**	Psycho-educational intervention on psychological outcomes	Usual care	484 (dyads)	Quality of life	Illness uncertainty	1
**Loucka 2024 [38]**	Primary palliative care intervention	Usual care	672	Quality of life	Hope, illness expectation	2

## Data Availability

Not applicable.

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
