# Peer review of "Assessment Tools to Examine Illness Understanding in Patients with Advanced Cancer: A Systematic Review of Randomized Clinical Trials"

_cancers, 2025, doi:10.3390/cancers17030385_

Round 1
Reviewer 1 Report
Comments and Suggestions for Authors
The manuscript describes a qualitative assessment of a literature review regarding end-of-life communication regarding illness understanding.
Recommendations are made to the authors to enhance the clarity of the presentation and the attached file.

Author Response
Comments 1: Create a separate table in the methods for line 32 in the abstract: “We extracted the questions, answers, definitions of accuracy, and accuracy rates of each category.”
Response 1: Thank you for your input. We have created a table of the 5 categories in the results section (Table 4). We will be happy to make further adjustments if needed.
Table 4. Summary of findings for each of the 5 assessment measurement categories. |
|||
Assessment measurement category |
Number of articles/RCT |
Number of unique assessments |
Accuracy rate (range) |
Prognostic Awareness |
10 articles, 8 RCTs |
6 |
6% to 33% |
Health Status |
4 articles, 3 RCTs |
1 overall main question, but answer choices varied slightly |
45% to 59% |
Curability |
9 articles, 6 RCTs |
5 |
35% to 84% |
Treatment Intent |
11 articles, 9 RCTs |
9 |
26% to 88% |
Treatment Risks/Benefits |
3 articles, 3 RCTs |
3 |
17% to 75% |
Comments 2: Indicate in the abstract (lines 33-35) as well as the results that only 18% of studies had a high quality JADA score of 3-5.
Response 2: Thank you for your feedback. We have now added the following in the abstract (p.1, paragraph 2): “The final sample included 27 articles based on data from 16 RCTs; five articles (18%) had a Jadad score of ≥ 3.”
Comments 3: Regarding Line 36 in abstract, create a short table of these findings in the results.
- Line 36 in abstract: “Only 4 RCTs examined illness understanding as a primary outcome or communication interventions.”
Response 3: Thank you for your comment. We have now created a table (Table 4) to summarize the overall findings for each of the 5 assessment measurement categories.
Table 4. Summary of findings for each of the 5 assessment measurement categories. |
|||
Assessment measurement category |
Number of articles/RCT |
Number of unique assessments |
Accuracy rate (range) |
Prognostic Awareness |
10 articles, 8 RCTs |
6 |
6% to 33% |
Health Status |
4 articles, 3 RCTs |
1 overall main question, but answer choices varied slightly |
45% to 59% |
Curability |
9 articles, 6 RCTs |
5 |
35% to 84% |
Treatment Intent |
11 articles, 9 RCTs |
9 |
26% to 88% |
Treatment Risks/Benefits |
3 articles, 3 RCTs |
3 |
17% to 75% |
Comments 4: In line 37-38 in the abstract, change to “…patient reported accuracy rate…”
Response 4: Thank you for your comment. We have now revised this to “accuracy rate of patients’ responses” in the abstract (p.1, paragraph 2).
Comments 5: In line 59 in the introduction, change to “…accuracy regarding cancer state…”
Response 5: Thank you for your comment. We have now revised this to “accuracy regarding cancer state” in the introduction (p.2, paragraph 2).
Comments 6: Regarding line 144-145 in Results section 3.2, list these 6 assessment methods in the text, and highlight them in table 3.
Response 6: Thank you for your comment. Based on your feedback, we have now highlighted and bolded each of the questions in Table 3 to make them clearly visible to the reader. Because there are many questions and some of them are quite long, we prefer to keep the questions in the table rather than the results text.
Comments 7: In Results section 3.3: Describe how the accuracy rate was determined – compared to physician estimates?
Response 7: Thank you for your comment. We have now added the following sentence in the methods for clarification (p.3, paragraph 5): “We also retrieved the accuracy rate of patients’ responses as defined by the authors in each study.”
Reviewer 2 Report
Comments and Suggestions for Authors
Cancers peer review
- The study is well designed and well written but lacks some of the foundational background and context to make its most important point. While I understand every intervention reviewed cannot be detailed due to the heterogeneity of the intervention types, there should be some inclusion of general intervention approaches in the background.
- Furthermore, awareness of terminal status is often a wavering construct with patients acknowledging their likelihood of death and the bouncing back into denial with hope or misunderstanding of provider-led education. This phenomenon is important to detail to help clinicians understand that checking for understanding may need to be a dynamic process with repeated check ins and review of previously given information as diseases progress. I would like to see this added to the intro as well as to the discussion.
- The tables are helpful but the articles reviewed need to be more clearly organized into domains such as: verbal interventions led by providers vs. structured interviews vs. questionnaires self-administered by patients
- The authors state that a limitation is that the surveys reflect baseline reports only. This is a major limitation given the wavering nature of EOL understanding but not one the authors had any control over given this is a meta analysis. I would like, nevertheless, to see this limitation discussed in terms of opportunities to improve clinical care. Because prognosis is not formally assessed in advanced cancer as disease progresses, how does this negatively impact decision making?
- It would also be good to highlight the ways in which academic medical centers, that have ongoing RCTs may actually have a conflict of interest in informing their patients of their death as they are simulataneously trying to recruit for trials and may therefore be confusing patients by contradicting themselves. For example, an oncologist may say: “we have no curative options left. But I have this study you can try.” This may confuse a patient from really grasping that survival is no longer on the table.
- The conclusions of the paper also mention the limitation of probability-based questions. In clinical practice, while providers are often asked to make guesses about length of time, this is often an educated guess that leaves chance for people to misinterpret survival. For example, when doctors say something like: 90% of patients like you don’t live past 2 years, many patients will naturally estimate themselves to be in the 10% that lives longer… but may not understand that this 10% may only live 1 extra year.
- Please address these larger themes throughout.
Author Response
Comments 1: The study is well designed and well written but lacks some of the foundational background and context to make its most important point. While I understand every intervention reviewed cannot be detailed due to the heterogeneity of the intervention types, there should be some inclusion of general intervention approaches in the background.
Response 1: Thank you for your comment. We have now added the following to the introduction (p. 2, paragraph 3): “We focused on randomized trials because we were most interested in how illness understanding was assessed as a variable (i.e., primary outcome, secondary outcome, co-variate, or baseline variable) in high quality studies, what the interventions were (e.g., communication tools, decision aids, or specialist palliative care), and if they specifically targeted illness understanding or other outcomes (e.g., quality of life).”
Comments 2: Furthermore, awareness of terminal status is often a wavering construct with patients acknowledging their likelihood of death and the bouncing back into denial with hope or misunderstanding of provider-led education. This phenomenon is important to detail to help clinicians understand that checking for understanding may need to be a dynamic process with repeated check ins and review of previously given information as diseases progress. I would like to see this added to the intro as well as to the discussion.
Response 2: Thank you for your comment. We have now added the following in the introduction (p. 1, paragraph 3): “Illness understanding may fluctuate over time.”
We have now also added the following to our discussion of limitations (p. 15, paragraph 5): “Fifth, because illness understanding fluctuates, longitudinal assessments are needed. Our systematic review is limited to what was reported and could not assess this in detail. However, these fluctuations may make it more difficult to detect signals in studies aiming to improve illness understanding and need to be further studied.”
Comments 3: The tables are helpful but the articles reviewed need to be more clearly organized into domains such as: verbal interventions led by providers vs. structured interviews vs. questionnaires self-administered by patients
Response 3: Thank you for your comment. While we agree with your suggestion, we were unable to examine the mode of survey administration because it was not clearly reported for each study. We have now added the following to our discussion of limitations (p. 15, paragraph 5): “Fourth, we were unable to examine the mode of survey administration because the survey administrator (i.e., the physician, research staff, patient) was not always clearly reported.”
Comments 4: The authors state that a limitation is that the surveys reflect baseline reports only. This is a major limitation given the wavering nature of EOL understanding but not one the authors had any control over given this is a meta analysis. I would like, nevertheless, to see this limitation discussed in terms of opportunities to improve clinical care. Because prognosis is not formally assessed in advanced cancer as disease progresses, how does this negatively impact decision making?
Response 4: Thank you for your comment. Based on your feedback, we have now added the following to our to our discussion of limitations (p. 15, paragraph 5): “Lastly, the majority of these studies examined illness understanding at baseline only and reported the findings as secondary analyses instead of being the primary outcome for a communication intervention. Given the high level of inaccuracy in illness understanding even at baseline, our study highlights the opportunity for systematic and routine screening in clinical settings to identify gaps in illness understanding and need for psycho-social-spiritual support.”
Comments 5: It would also be good to highlight the ways in which academic medical centers, that have ongoing RCTs may actually have a conflict of interest in informing their patients of their death as they are simulataneously trying to recruit for trials and may therefore be confusing patients by contradicting themselves. For example, an oncologist may say: “we have no curative options left. But I have this study you can try.” This may confuse a patient from really grasping that survival is no longer on the table.
Response 5: Thank you for your comment. Based on your feedback, we have now added the following to our discussion of limitations (p. 15, paragraph 5): “Because our study focused on RCTs, which are typically done at academic centers that serve a different patient population compared to smaller hospitals, further studies are needed to examine patient illness understanding in other settings.”
Comments 6: The conclusions of the paper also mention the limitation of probability-based questions. In clinical practice, while providers are often asked to make guesses about length of time, this is often an educated guess that leaves chance for people to misinterpret survival. For example, when doctors say something like: 90% of patients like you don’t live past 2 years, many patients will naturally estimate themselves to be in the 10% that lives longer… but may not understand that this 10% may only live 1 extra year.
Response 6: Thank you for your insights. We agree this is another opportunity for reflection and have added the following in the discussion section (p. 16, paragraph 2): “Probability-based questions are also harder to code as accurate or inaccurate. In addition, such questions may be interpreted differently depending on each individual’s outlook (i.e., positive or negative) and numeracy.”
Comments 7: Please address these larger themes throughout.
Response 7: Thank you for your suggestion. We have reviewed your comments and adjusted the text accordingly throughout the manuscript.